# The Impact of Reflection on Death on the Self-Esteem of Health Care Workers

**DOI:** 10.3390/ijerph19095521

**Published:** 2022-05-02

**Authors:** Piotr Holajn, Agata Zdun-Ryżewska, Marlena Robakowska, Daniel Ślęzak, Anna Tyrańska-Fobke, Andrzej Basiński

**Affiliations:** 1Department of Medical Rescue, Medical University of Gdańsk, 80-210 Gdańsk, Poland; plhol@gumed.edu.pl (P.H.); andrzej.basinski@gumed.edu.pl (A.B.); 2Department of Life Quality Research, Medical University of Gdańsk, 80-210 Gdańsk, Poland; agata.zdun-ryzewska@gumed.edu.pl; 3Department of Public Health and Social Medicine, Medical University of Gdańsk, 80-210 Gdańsk, Poland; mrobakowska@gumed.edu.pl

**Keywords:** self-esteem, communion, agency, TMT, mood, religious attitude

## Abstract

Background. The study was based on the Terror Management Theory. This theory assumes that self-preservation and awareness of imminent death create the potential to trigger fear. The “culture buffer” can protect people from fear, and it is composed of two factors: personal views on world issues and self-esteem. The aim of the study was to show that exposure to content that increases the availability of thoughts about death causes changes in medical personnel (doctors, nurses, and paramedics) in areas such as self-esteem, mood, sense of agency, and communion. Methods. The research was experimental. Standardized psychometric tests were used, including the Rosenberg self-esteem scale (RSE), the University of Wales Institution of Science and Technology) Mood Adjective Check List (UMACL), scales measuring agency and communion, and an additional questionnaire containing two types of text. Respondents were divided into two text groups: A (exposed to increased availability of thoughts of death) and B (neutral). Results. Reflection on death, triggered by the experimental manipulation of the independent variable (text version), did not modify mood (in groups of medical staff and students) or self-esteem of health care professionals but did modify scores on a single RSE item in the student’s group. Moreover, age, income level, religious attitude, and belonging to a professional group had an impact on self-esteem, mood components, and other parameters but did not interact with the text group. Reflection on death modified the sense of agency and communion. Conclusions. Exposure to content increasing the availability of thoughts of death led to observable effects possible to observe in all groups only after taking into account an additional factor, which turned out to be the religious attitude of the respondents in the experiment. Specific tools should be selected or developed for the needs of research on respondents working in health care.

## 1. Introduction

The Terror Management Theory (TMT), developed by Solomon, Greenberg, and Pyszczynski [1], is based on a concept by Ernest Becker [2]. The TMT attempts to propose a concept of human behavior and motivations from the socio-biological perspective and is based upon two assumptions. Firstly, the observation that in a life-threatening situation, through the abundance of thoughts about death, an emotional reaction, called ‘fear’, appears. Such situations include witnessing, participating in, or even envisaging events that may lead to fatalities, such as catastrophes, disasters, and other problems, whether they be collective or individual [3]. Secondly, the assumption that there exists a culture buffer, acting as protection against the fear of death, and triggered by intensified thoughts or images related to death. The buffer consists of two factors—personal worldview and self-esteem. The fear of death can be controlled by assuming a certain worldview, which creates for the individual an illusion of understanding of reality, thus evoking a sense of order, meaning, and stability and providing guidelines for good practices and behavior. The TMT defines self-esteem as a set of notions about oneself in relation to social and cultural norms. A subjective belief in being able to control events and own agency increases the level of self-esteem [4]. Crucially, both self-esteem and worldview need external validation and feedback to reaffirm their validity and justify their meaning within a specific culture. The more people adopt a given worldview, the more efficient the cultural buffer is. Likewise, if self-esteem is damaged, one looks to find the means of restoring it [5]. Summing up, the hypothesis of the cultural buffer assumes that increased pressure on the buffer results in weakening self-esteem and increasing fear, whereas a decreased pressure on the buffer results in strengthening self-esteem and decreasing fear. The fear of death is not a constant. It is evoked and reinforced by the reflection on death that arises, e.g., at the moment of direct contact with the dying, a situation to which health care workers are obviously exposed.

The main aim of the research was to determine whether contact with the dying and reflection on death influenced self-esteem and mood, as well as the worldview, including beliefs regarding agency and communion among medical staff and students. An attempt was also made to verify whether additional factors, such as religious attitudes and socio-demographic variables, had an impact on the tested relationship.

## 2. Materials and Methods

### 2.1. Study Group

The research was carried out in 2019 on a group of 200 persons, including 113 representatives of medical staff: doctors, nurses, and paramedics, as well as 87 last-year students of medical faculties related to one academic and clinical center.

The medical staff group consisted of 19 doctors (13 men and 6 women), 52 nurses (5 men and 47 women), and 42 paramedics (32 men and 10 women). The last-year student group consisted of 15 students of medicine (9 men and 6 women), 40 students of nursing (4 men and 36 women), and 32 students of paramedics (16 men and 16 women). The whole sample consisted of 76 men and 124 women, with the youngest respondent 21 years old and the oldest 63 years old, and a mean age of 39 years old. 

### 2.2. Methods

An experiment was carried out to show that displaying content that increases the availability of thoughts about death causes a change in the activity of medical personnel (doctors, nurses, paramedics) in such areas as: self-esteem, mood, and worldview (including agency and communion). The results were compared to the groups of students in the last year of studies in medical faculties matched with the professional groups.

The independent variable (death/no-death text) underwent experimental manipulation consisting of evoking thoughts of death by using two versions of a primary questionnaire completed by the respondents. Two versions of the questionnaire were created for the purpose of the experiment. 

One of the primary questionnaires (version A) was designed to elicit thoughts of death, while the second primary questionnaire (version B) remained neutral.

Both questionnaires contained general lifestyle and socio-demographic questions, which, in version A, invoked thoughts of death, e.g., ‘Are your parents alive?’, ‘Are all you children alive?’). A death-related fragment of ‘Bridge to Terabitha by Katherine Parkinson, translated by Alicja Skarbińska [6], was also included. Version B contained questions on spending leisure time, opinions about classical music, and a fragment of ‘Bridge to Terabitha’ with neutral content. 

In its main part, the study utilized the following standardized psychometric tests: Rosenberg’s self-esteem scale (RSE), in the Polish adaptation by Mariola Łaguna, Kinga Lachowicz-Tabaczek and Irena Dzwonkowska [7]; University of Wales Institution of Science and Technology Mood Adjective Check List (G. Mathews, A.G. Chamberlain and D.M. Jones, in the Polish adaptation by Ewa Goryńska -UMACL [8]); and scales measuring agency and communion by Wojciszke, B. and Szlendak, M. [9] as well as a customized questionnaire containing statements about group work, using a framework created by Lencioni, dealing with five dysfunctions of group work and three statements regarding self-destructive behaviors, often described as inefficient stress management.

The RSE consists of ten items. It allows for measuring general self-esteem levels; thus, the personal attitude to one’s self, demonstrated by a self-description, is treated as a highly constant characteristic, not a temporary state [6]. The scale is a one-dimensional tool. It is comprised of ten diagnostic statements. The respondent is asked to mark, on a four-point scale, how far they agree with each of the ten statements. The results are highly reliable—the Cronbach’s alpha indicators for different age groups vary from 0.81 to 0.83 [7].

UMACL (University of Wales Institution of Science and Technology) Mood Adjective Check List),) is a scale for use with adult and adolescent subjects, consisting of a spreadsheet with twenty-nine adjectives. The standard procedure is to give an answer on how far each of the adjectives reflects the current mood of the respondent (on a four-point scale, ranging from “definitely yes” to “definitely not”). The checklist is completed individually or in a group, with no fixed time limit. The scale measures the current (immediate) mood, defined as an affective experience of medium length (at least a few minutes long), not related to an object or related to a quasi-object, involving three dimensions of basic affective state: Tense Arousal (TA), Hedonic Tone (HT) and Energetic Arousal (EA) [8]. 

The scales measuring agency and communion [9] can be used to measure unmitigated agency and unmitigated communion and are suitable for application in adults and adolescents individually or in a group. They can be used to measure agency orientation (defined as focusing on one’s self and one’s goals), communion orientation (defined as focusing on other people and interpersonal relations), unmitigated agency (excessive focus on one’s self coupled with ignoring social relations) and unmitigated communion (excessive focus on others coupled with ignoring one’s agency). The scales measuring agency and communion consist of 30 items (single adjectives or phrases) denoting traits, and the unmitigated agency and communion scales consist of 22 items in the form of statements. The answers are given on a 7-point scale. Assessing the reliability in both scales demonstrated high internal consistency of the scales: Cronbach’s alpha 0.92 for communion and Cronbach’s alpha 0.90 for agency. In the case of unmitigated communion and unmitigated agency, moderately high indicators of internal consistency were obtained: Cronbach’s alpha 0.86 for unmitigated communion and Cronbach’s alpha 0.73 for unmitigated agency [9].

Research preparation: the research was carried out at the workplace or educational facility of the tested group by one tester, which allowed to eliminate any influence of the tester on the participants. The participants verbally agreed to take part in the test, were informed about the purpose of the study, and reassured of the anonymity of the results.

Due to the lack of differences in the results between RES and UMACL results obtained in an individual and group conditions, and the lack of information on the impact of the test method (individually or in a group) on the Orientation Measurement Scale, as described in the literature, some measurements were completed in small groups of several people. Paper questionnaires in the mother tongue of the respondents, i.e., in Polish, were completed by the respondents.

### 2.3. Methodology

In order to test the research hypotheses, statistical analyses were carried out using the IBM SPSS Statistics 25 program [10]. The analysis included basic descriptive statistics, the Shapiro–Wilk and Kolmogorov–Smirnov distribution normality tests, Student’s *t* tests for independent samples, and correlation analysis using r (Pearson) and rho (Spearman) coefficients. Two-way ANOVA and one-way ANOVA for independent samples were also performed. The *p* < 0.05 was assumed as the level of statistical significance.

## 3. Results

### 3.1. Self-Esteem

#### 3.1.1. Provoking Thoughts about Death: The Impact of Text Materials on Self-Esteem

In order to verify whether provoking thoughts about death has a statistically significant influence on the dependent variables measured in the study, several Student’s *t*-tests for independent samples were conducted. The analyses were conducted separately for two study groups. The group of students was the control group. In the first part, an analysis of scores for items included in the self-esteem scale was carried out. Statistically significant differences were observed regarding the RSE item ‘On the whole I’m satisfied with myself’. The tested students who were given version A of the text demonstrated a higher mean score in this respect when compared with students given version B of the text, which means that version A students were indeed overall happier with themselves. Moreover, a statistically significant result was observed regarding the ‘I feel I do not have much to be proud of’ item. For this item, version B students had a higher average score; however, the differences between students in the version A and version B groups were not as big. No other significant differences regarding self-esteem, and its items, were observed. In the group of medical staff, activating thoughts of death did not influence the self-esteem measures. In this instance, the Student’s t-test was proved statistically not significant.

#### 3.1.2. The Influence of the Thoughts about Death and Religious Attitudes on Self-Esteem

Next, a two-factor-between-groups variance analysis was completed. The analysis included the whole study group (students and medical staff), with the text (A, invoking-death-reflection vs. B not-invoking-death-reflection) and religious attitude treated as the independent variables. The analysis demonstrated a significant interaction regarding the self-esteem item ‘I certainly feel useless at times.’, F (1; 81) = 4.02; *p* < 0.05; η^2^ = 0.05. Pair comparisons showed significant differences in scores on this item depending on text version and religious attitudes. Although no main effects of religious attitude or text were observed for any of the dependent variables, an interaction between religious attitude and experimental manipulation was shown for “I certainly feel useless at times” SES item, with non-believers achieving significantly lower scores for this item (indicating increased self-esteem) when presented with text A (death-reflection-invoking) than to text B (neutral) (Table 1). 

### 3.2. Mood

#### 3.2.1. Provoking Thoughts about Death: The Impact of Text Materials on Mood

The next step, also using the Student’s t-test for independent samples, involved checking whether increasing the accessibility of thoughts of death had had any influence on the participants’ mood. There were no statistically significant differences between versions A and B of the text, therefore proving that an increase in the availability of thoughts of death had no influence on the mood among the students and medical staff who took the test.

#### 3.2.2. Relationship between Age and Mood Depending on the Text Group

The Pearson’s r coefficient correlation analysis examined whether age (in the whole group of respondents) was related to mood, depending on the availability of thoughts about death. Both among text A and text B respondents, age correlated with Energetic Arousal, and these correlations were statistically significant. This correlation was positive, with Energetic Arousal higher with higher age (Table 2).

#### 3.2.3. The Relationship between Income Level and Mood Depending on the Text Group

Spearman’s rho coefficient analysis was used to check the relationship between income level and mood. The whole study group was analyzed (both students and medical staff). Significant correlations, positive albeit weak, were obtained in the group presented with text A regarding Hedonic Tone. On the other hand, in the group presented with text B, the level of income correlated statistically significantly with Tense Arousal. These were negative and weak correlations, so an increase in the income level of text B respondents results in a decrease in their level of Tense Arousal (Table 3).

### 3.3. Agency and Communion

#### 3.3.1. Provoking Thoughts about Death: The Impact of Text Materials on Agency and Communion

The next Student’s t-test analysis for independent samples was aimed at checking whether increased availability of thoughts of death had an influence on oriented agency and oriented communion. A statistically insignificant result was obtained in each of the groups, thus proving that the students and medical staff presented with either version A or B did not show any significant differences regarding oriented agency and oriented communion.

#### 3.3.2. The Influence of Religious Attitudes on the Agency and Communion Orientation

In the case of the relationship between religious belief and agency and communion orientation among students, statistically significant differences in terms of sense of agency, communion, and unmitigated communion were observed. The average scores show that students who declared themselves as believers were characterized by a significantly higher level of agency as well as communion and unmitigated communion when compared to non-believers. Unmitigated agency differences were moderate, communion differences were strong, and unmitigated communion differences were very strong. 

In the group of medical staff, statistically significant differences were noted in the sense of communion, unmitigated agency, and communion. It turns out that believers were characterized by a significantly higher level of communion and unmitigated communion, as well as a lower level of unmitigated agency compared to medical staff members who identified as non-believers. The differences between unmitigated agency were moderately strong, communion differences were strong, and unmitigated communion differences were very strong (Table 4). 

In order to verify whether the text group and religious attitude had a statistically significant influence on the agency and communion orientation, a number of two-factor analyses of variance were carried out separately for students and employees. As a result of the analyses, a trend toward both factors influencing the measure of unmitigated communion in the group of students was established—F (1; 81) = 3.38; *p* = 0.070; η^2^ = 0.04. Post hoc tests showed that non-believers presented with text A differed statistically significantly (*p* = 0.037) from non-believers presented with text B, obtaining a lower result on the scale of unmitigated communion. In the believers’ group, the text did not differentiate the respondents. Moreover, statistically significant differences were observed between believers and non-believers among the respondents with text A (*p* = 0.006). Believers in this group were characterized by a significantly higher level of unmitigated communion than non-believers.

In the group of doctors, the effect of the text and religious attitude interaction was significant at the level of the statistical tendency for the dependent variable ‘sense of agency’—F (1; 14) = 3.21; *p* = 0.095; η^2^ = 0.19. According to the post hoc tests, non-believers presented with text A differed from non-believers presented with text B, obtaining a much higher result in terms of the sense of agency (*p* = 0.036). Moreover, at the level of statistical tendency (*p* = 0.061), non-believers differed significantly from believers, but only when they received text B. Believers with text B have a higher level of agency than non-believers with the same text.

In the group of nurses, a statistically significant effect of the interaction of the text group and religious attitude was obtained for the dependent variable ‘sense of communion’—F (1; 47) = 5.50; *p* = 0.023; η^2^ = 0.11. Post hoc tests proved that non-believers presented with text A differed from non-believers presented with text B, obtaining a much higher result in terms of the sense of communion (*p* = 0.017). On the other hand, non-believers differed significantly (*p* = 0.001) from believers, but only when the A version of the text was presented to them. Believers presented with text A were characterized by a higher level of communion than non-believers.

#### 3.3.3. The Correlation between Income Level and Agency and Communion Orientation Depending on the Text Group

Spearman’s rho coefficient was used to analyze whether the level of income was related to the agency and communion orientations. Significant relationships were obtained in the whole study group (medical staff and students) presented with text A regarding the sense of agency. These correlations had a positive sign and low strength, which means that the higher the income of the respondents from this group, the higher their agency scores. On the other hand, in the group presented with text B, the level of income correlated statistically significantly with the feeling of unmitigated communion. These were negative and weak correlations; therefore, among text B respondents, a higher income was correlated with lower levels of unmitigated communion. This was the case for the whole study group, without a division into medical staff and students (Table 5).

## 4. Discussion

This paper was based on the assumption that medical staff have direct contact with emergency rescue patients, which triggers a reflection on one’s mortality. As proposed by TMT, a buffer is triggered in these situations. Research evidence suggests that encounters undermining the worldview values stimulate the activity aimed at “defending” the individual from the results of their reflection on death. The buffer alters the human functioning in various areas, for example, concerning a sense of agency and communion-oriented activities. Professionals employed by the health service, in direct contact with the dying, being at the same time consumers, and even co-creators, of mass culture, are susceptible to permanent cognitive dissonance [11]. These issues have been acknowledged by researchers. However, the papers written so far have focused on effects such as burnout [12] or PTSD [13]. Tools aimed at measuring the damaging effect of stress [14] and burnout [15] have been established. 

Given the TMT, it was assumed that effects obtained among our research group would be similar to those obtained during the TMT creator’s experiments [5], as well as the derived conclusions [16].

The main aim of our study was to verify whether medical staff exposed to text A (death-reflection-invoking) triggered a buffer, resulting in higher self-esteem, changes in mood, agency, and communion. The occurrence of differences after exposure to text A would justify a statement that an increase in thoughts of death led to higher self-esteem. The results of our study do not allow for a claim of such effect among medical staff and students of medical faculties. A detailed analysis of Rosenberg’s Self Esteem Scale—RSE— showed that there was a statistically significant difference regarding the positive statement ‘On the whole I’m satisfied with myself’. The result means that students presented with a text increasing the availability of thoughts of death were happier with themselves (*p* < 0.041), which is consistent with the TMT.

Such an effect was not observed in the medical staff group, which might support the TMT creators’ assumptions that, in their case, self-esteem is a much more stable construct, enabling a suppressing mechanism to hold back the increasing number of thoughts about one’s mortality. What is more, an opposite tendency to this was observed among group B students. The statistical tendency to agree with the statement ‘I feel I do not have much to be proud of.’ appeared in this instance, which demonstrates a negative impact (*p* < 0.87). Despite numerous analyses of the collected data, neither an influence on the mood of the tested group was recorded, nor was any difference in the TA, EA, or HT measures observed in any of the tested groups. In addition, no statistically significant difference resulting from exposure to text A or B, with regard to agency, communion, and the unmitigated versions thereof, was visible. 

One possible explanation lies in the specifics of the professions of the tested group members, which require particular personal characteristics, further enhanced during professional experience and socializing with other members of the profession. These skills include, to name a few, the ability to cope with stress and to work in difficult situations requiring quick decision-making while under high and sometimes extreme pressure [16]. It is also possible that a skillset such as this is already present among the students of medical faculties, which, given the above-mentioned working environment, is even more likely to develop. It would explain the occurrence of weak yet statistically significant differences and the existence of statistical tendencies visible only in the students’ group [17]. 

Religious attitudes/beliefs, as a factor strongly influencing the worldview, was included in the experimental plan as a factor that would potentially modify the influence of activating thoughts of death in the participants as it would also allow for the effective defense. 

This allowed for the formulation of conclusions that the hypothesis of the salience of death concerns not those who fear death because it can be assumed that everyone is afraid of death, “but those who (…) fear God”. In the respondents who declared themselves as religious, it was much easier to observe the effects of the salience of death caused by the experimental manipulation [3].

For the purposes of the study, the respondents in all groups were asked whether they were believers. The study was conducted on a group of 200 people, 153 of whom declared themselves as religious. The question of belonging to a particular church or religious association was not specified. There was, however, a place in the questionnaire to describe the religion. As it turned out, despite the lack of influence of religiousness on self-esteem or mood, religiousness was statistically significant for the sense of agency and communion. It is necessary to note at this point that the subjective belief in the ability to control actions and the sense of security resulting from belonging to a group are important elements that increase self-esteem [18].

In the study, it was shown that students identifying themselves as believers were characterized by a significantly higher level of agency and communion, as well as unmitigated communion when compared to non-believer students. What is more, statistically significant differences were observed among the medical staff, where believers were characterized by a higher level of communion and unmitigated communion as well as a lower level of unmitigated agency.

These results were the base for further analyses, including religious attitudes and a given text group, as well as the joint impact of these factors on agency. Several two-factor variance analyses of both students and staff were completed.

However, post hoc tests showed that non-believers (students and staff together) in the A text group obtained statistically significantly different unmitigated communion scores (*p* = 0.037) compared to non-believers in the text B group. What is surprising is that if references to the perceived communion were used to buffer the fear of death, one could expect a higher rather than a lower score on this scale. This may indicate that there are additional factors that have an impact on the buffering of death anxiety, which were not captured in this study.

Moreover, statistically significant differences were observed between believers and non-believers among the respondents presented with text A (*p* = 0.006). Believers in this group were characterized by a significantly higher level of unmitigated communion than non-believers. Such a result can be recognized as consistent with the assumptions of the Terror Management Theory, assuming that the reflection caused by the availability of thoughts about death causes changes not in the area of self-esteem but in the sense of agency and communion, which can be considered as components of self-esteem [17]. 

Post hoc tests proved that non-believers presented with text A differed from non-believer text B respondents with respect to communion (*p* = 0.017). One could state that this was due to the fact that non-believers had to refer to values other than religion. This test proved that, indeed, non-believer respondents, upon intensified exposure to content increasing the availability of thoughts of death, experience and share the values of the group they belong to. A direct influence of religion on self-esteem was not observed. An additional two-factor analysis of the influence of the text and religion on self-esteem was conducted, though. With regards to the RSE scale item, ‘I certainly feel useless at times.’ it was shown that non-believer respondents who were not exposed to increased availability of thoughts of death did indeed think of themselves as useless far more often than non-believers presented with text A. (F (1; 81) = 4.02; *p* < 0.05; η^2^ = 0.05). This indicates that the former did not find support in religion. Therefore, the absence of religion prevents the triggering of the buffer and results in lower self-esteem for this item on the RSE scale.

During the analysis of the influence of religious attitude on agency and communion, it was possible to demonstrate that students who declared themselves as believers were characterized by a significantly higher level of agency as well as unmitigated communion and communion compared to the non-believer students. Moreover, statistically significant differences were also noted in the group of medical staff. The believers from this group were characterized by a significantly higher level of community and unmitigated communion and a lower sense of unmitigated agency.

Correlation analysis using the Pearson r coefficient was used to verify whether age correlates with mood among the surveyed students and medical staff. It appears that among students, age was statistically significantly correlated with Tense Arousal. It was a weak and positive correlation; therefore, the older the students were, the stronger their Tense Arousal was. In turn, among the employed, a relationship between age and Energetic Arousal was observed at the level of statistical tendency. This relationship was also weak and positive, so as the age of the respondents in this group increased, their level of Energetic Arousal increased too.

It turned out that age correlated at the level of statistical tendency with the sense of agency only in the group of the employed. It is a negative and weak correlation, which means that the older the working respondents were, the weaker their sense of agency was. There were, however, no other trends in the relationships between age and agency and communion.

It is possible that the inconclusive results of the study were due to a weak and thus not entirely successful experimental manipulation. However, exposing the respondents to stronger stimuli such as for example graphic photographs associated with death might be considered unethical. Conducting such an experiment would have to involve psychological assistance, which would need to be provided to the participants in order to minimize the negative effects following the exposure [19]. 

The size of the groups may also be considered to be one of the limitations of the study. However, despite this, the main assumptions were confirmed in this study. It is possible to continue the research focused on a larger group of specialists and students in the future.

## 5. Conclusions

The cultural buffer that is triggered in the distal defense of the fear of death arises not only on the basis of modifications in the scope of self-esteem. The construction of the Terror Management Theory allows seeing other elements influencing the activation of the buffer and thus directs the interest to other research areas.

Reflection on death, triggered by the experimental manipulation of the independent variable, did not modify the self-esteem scores of health care professionals in the area directly related to saving lives. This reflection was triggered in the group of students and only modified scores on individual items from which the RSE scale is constructed.Reflection on death did not affect the mood or any of its components in the studied groups. Various states of the components of the mood, i.e., TA, EA, or HT, were related to factors other than experimental manipulation of death reflection.Reflection on death modified the sense of agency and community of professionals who deal with the problem of death in their professional practice. Changes in this respect were possible to observe in all groups only after taking into account an additional factor, which turned out to be the religious attitude of the respondents in the experiment.Other factors, such as age, length of service, income level, and belonging to a specific professional group, affected self-esteem and mood components, but they did not interact with exposure to content that increased the availability of thoughts about death.

There is a unique specificity of the group covered by this study, and this difference should be taken into account when planning further research. Specific tools should be selected or developed for the needs of research concerning professionals employed in healthcare, and research should be conducted by interdisciplinary teams composed of psychologists cooperating with healthcare professionals.

## Figures and Tables

**Table 1 ijerph-19-05521-t001:** Descriptive statistics for the dependent variable RSE item depending on the text group and religiosity.

Dependent Variable:				
Sometimes I Certainly Feel Useless at Times.				
A/B text	Religious Attitude	M	SD	*n*
Text B	Non-believer	3.15	0.80	13.00
	Believer	2.86	0.71	28.00
	Overall	2.95	0.74	41.00
Text A	Non-believer	2.67	0.49	12.00
	Believer	3.03	0.69	32.00
	Overall	2.93	0.66	44.00
Overall	Non-believer	2.92	0.70	25.00
	Believer	2.95	0.70	60.00
	Overall	2.94	0.70	85.00

**Table 2 ijerph-19-05521-t002:** Analysis of the correlation between age and mood depending on the group.

		Age	
		Text A	Text B
Tense Arousal (TA)	Pearson’s r	−0.132	−0.079
	Significance	0.175	0.450
Energetic Arousal (EA)	Pearson’s r	0.243	0.289
	Significance	0.012	0.005
Hedonic Tone (HT)	Pearson’s r	0.076	0.065
	Significance	0.435	0.538

**Table 3 ijerph-19-05521-t003:** Analysis of the correlation between the income level and mood depending on the text group.

		Income Level	
		Text A	Text B
Tense Arousal (TA)	Spearman’s Rho	−0.112	−0.279
	Significance	0.272	0.014
Energetic Arousal (EA)	Spearman’s Rho	0.090	0.088
	Significance	0.376	0.448
Hedonic Tone (HT)	Spearman’s Rho	0.196	0.182
	Significance	0.053	0.116

**Table 4 ijerph-19-05521-t004:** Differences in agency and communion orientation in the group of medical staff, depending on religious attitudes.

	Non-Believers (*n* = 19)		Believers (*n* = 93)				95% CI		
	M.	SD	M.	SD	vol	*p*	LL	UL	d Cohen
A sense of agency	5.11	0.86	5.35	0.78	−1.23	0.222	−0.64	0.15	0.31
A sense of communion	5.20	0.90	5.65	0.64	−2.57	0.011	−0.79	−0.10	0.65
A sense of unmitigated agency	4.08	0.80	3.67	0.81	2.01	0.047	0.01	0.81	0.51
A sense of unmitigated communion	3.81	0.86	4.52	0.86	−3.30	0.001	−1.15	−0.29	0.83

M—mean; SD—standard deviation; *p*—significance level; CI—confidence interval; LL, UL—lower, upper limit of CI.

**Table 5 ijerph-19-05521-t005:** Analysis of the correlation between the level of income and agency and communion orientation depending on the text group.

		Income Level	
		Text A	Text B
Sense of agency	Spearman’s Rho	0.237	0.161
	Significance	0.019	0.165
Sense of communion	Spearman’s Rho	−0.043	−0.077
	Significance	0.678	0.506
Sense of unmitigated agency	Spearman’s Rho	0.019	−0.050
	Significance	0.849	0.669
Sense of unmitigated communion	Spearman’s Rho	−0.019	−0.281
	Significance	0.854	0.014

## Data Availability

Data sharing not applicable.

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
