# Peer review of "The Impact of Reflection on Death on the Self-Esteem of Health Care Workers"

_ijerph, 2022, doi:10.3390/ijerph19095521_

Round 1
Reviewer 1 Report
Dear Authors,
In general, the manuscript has a much clearer introduction, in which the relationships between Terror Management Theory, agentivity, self-esteem, mood and sense of community finally emerge.
The method is also much improved and full of important details. The sample is now described in detail. Great job.
The results are still unclear, so I recommend re-reading and proceeding in the same order in investigating the hypotheses and instruments. E.g. 1. self-esteem, 2. mood, 3. sense of community. Try to simplify, to clarify.
It is not always clear in the results section which groups are being compared (students vs. workers? high and low self-esteem?).
On the other hand, the discussions are adequate and underline the importance of the topic and that reflection on death has not so much an impact on mood, but on self-esteem and communion among health workers, which makes this study of great value.
Best regards
Author Response
Thank you. We chcanged the result section and added required details. We established that section in suggested order: 1. self-esteem, 2. mood, 3. sense of community. In addition, we also revised the entire manuscript linguistically.This manuscript is a resubmission of an earlier submission. The following is a list of the peer review reports and author responses from that submission.
Round 1
Reviewer 1 Report
The study deals with an interesting topic, very relevant to the medical profession and often underestimated.
I have just a few suggestions/comments
- It is not specified whether the authors intend to investigate the sense of self-esteem as an individual (i.e. I am a good person) or professional self-esteem, e.g. feeling effective in caring for a patient (since they selected doctors and nurses). And It is not directly clear how reflection on death can be related to self-esteem. Some more explanation in the introductory section would be helpful.
- In the "study group" it is not specified whether the subjects are workers belonging to one or different hospital centres, how many males and females, minimum, maximum and average age. It is not specified in which year data were collected, e.g. would it be different if collected before or after the covid-19 pandemic (in terms of death exposure)?
- The method section is too concise when describing the tools. For the reader is not clear the structure of the questionnaires, as emerges in the tables. For example, the following details are not specified: number of items the questionnaire consists of, sub-dimensions, sub-scales, calculation of the score etc.
- In which language were the reading text and the questionnaires administered? Please specify. Also, was the administration in groups or individual? Was it in person or online?
- In the discussions at least from line 219 to line 278 the authors describe results with significance indices and numerical values. It would be more appropriate to separate the numerical Results from the scientific comments and reflections (Discussion).
- The bibliography is very limited. Several bibliographical references are missing. May I suggest adding:
- An original reference for Terror Management Theory (TMT).
- A bibliographical reference for Rosenberg's Self-Esteem Scale (SES)
- Bibliographic reference of the Adjective Mood Scale (UMACL)
- Bibliographic reference of the Scale for measuring agency and community
- IBM SPSS software reference
Author Response
Thank you for all suggestions. Please see the attachment.

Reviewer 2 Report
I don't see the interest of this article and I do not understand the objective and the practical relevance of the study. I think the conclusions are obvious.
1. What is the main question addressed by the research?
The impact of reflection on death on the self-esteem of health care workers, but this reflection does not have an objective, it is an, in my opinion, irrelevant and obvious feature. It is clear that the thought of death has an impact on people and in particular, on medical and health workers. But what is the objective of this observation?
2. Do you consider the topic original or relevant in the field, and if so, why?
Not from my perspective, because there is no reference to health and safety at work, which, by the way, is already known and analized
3. What does it add to the subject area compared with other published material?
Nothing
4. What specific improvements could the authors consider regarding the methodology?
basis, purpose, ground
5. Are the conclusions consistent with the evidence and arguments presented and do they address the main question posed?
Yes, because there is a logical correlation
6. Are the references appropriate?
In my opinion, there are very few
7. Please include any additional comments on the tables and figures.
I do not have any comment on it
Author Response

(The authors gave the same response as above.)
